# Debiased Medical Report Generation with High-Frequency Amplification

## Abstract

In recent years, automated medical report generation (MRG) has gained significant research value for its potential to reduce workload and prevent diagnostic errors. However, generating accurate radiology reports remains challenging due to the prevalence of normal regions in X-ray images and normal descriptions in medical reports. Despite various efforts to address these issues, the definitions of *visual bias* and *textual bias* remain unclear and there is still a lack of comprehensive analysis of how these biases affect model behavior. In this work, we rigorously define and conduct an in-depth examination of visual and textual biases inherent in MRG datasets. Our analysis emphasizes that global patterns, such as normal regions and findings, contribute to visual and textual bias. Further, we discuss how these biases make MRG models especially prone to frequency bias, where models tend to prioritize low-frequency signals that capture global patterns, while neglecting high-frequency signals. To debiase the frequency bias, we propose the *high-frequency amplification layer* (HAL), aimed at enhancing the model's perceptiveness to fine-grained details. Our extensive experiments show that by amplifying high-frequency signals, HAL reduces both visual and textual biases, leading to improved performance in MRG tasks.

## 1 Introduction

The automation of diagnosis and treatment using medical images has received growing attention in both academia and industry (Wolleb et al., 2022; Manzari et al., 2023; Jiang et al., 2022). In particular, medical report generation (MRG) is one of the most promising tasks as it can alleviate the heavy burden of radiologists and reduce diagnostic errors. MRG aims to automatically generate a free-text description given a medical image (e.g., chest X-ray), describing the detailed findings on both normal and abnormal regions.

Generating diagnostically accurate and domain-specific radiology reports is challenging due to the presence of severe visual and textual biases. From the perspective of data, most medical images are dominated by normal regions, making it difficult to capture distinct features (see Figure 1a). Similarly, medical reports primarily describe normal findings, complicating the explanation of abnormal findings (see Figure 1b). Recently, several methods have been proposed to address visual and textual biases (You et al., 2021; Liu et al., 2021a; Tanida et al., 2023; Zhang et al., 2020; Liu et al., 2021a; Huang et al., 2023; Li et al., 2023). However, the definitions of visual bias and textual bias have not been clearly established and there remains a lack of comprehensive analytical understanding of how these biases affect model behavior.

Our work focuses on rigorously defining and identifying the fundamental challenges in MRG, analyzing how visual and textual biases hinder model performance. Further, from the perspective of the model, we relate these biases to frequency bias, where the model tends to capture low-frequency signals, while neglecting high-frequency signals. In this context, we associate normal features with low-frequency signals and abnormal features with high-frequency signals. In MRG, where transformers are widely used, this issue is exacerbated by inherent visual and textual biases. To address this fundamental challenge, we introduce a simple method called high-frequency amplification, which amplifies high-frequency signals to better capture abnormal features. We demonstrate that this simple approach effectively debiases frequency bias through extensive experiments, including pseudo-spectrogram analysis, cross-attention analysis, and representation analysis. We evaluate our

model on two benchmarks, MIMIC-CXR (Johnson et al., 2019) and IU X-ray (Demner-Fushman et al., 2016). The contributions of our study can be summarized as follows:

- We precisely define visual bias and textual bias, which are crucial but underexplored challenges in MRG. Through comprehensive analysis, we empirically confirm the presence of each bias and show how they exacerbate frequency bias. We emphasize that debiasing and mitigating frequency bias is a fundamental challenge that must be addressed in MRG tasks.

- We introduce a simple yet effective method named high-frequency amplification, specifically designed to mitigate the dominance of normal features in medical images and reports. By amplifying high-frequency signals, which correspond to abnormal features, our approach enables models to effectively capture both global and local patterns.

- We validate the effectiveness of our approach through extensive experiments, including pseudo-spectrogram analysis, cross-attention analysis, and representation analysis. We demonstrate our simple approach achieves performance superior or comparable to state-of-the-art models across both natural language generation and clinical efficacy metrics.

## 2 RELATED WORKS

Most existing MRG methods follow standard image captioning approaches due to the similarities between the two tasks. Despite remarkable success in image captioning models, MRG still faces significant challenges due to severe visual and textual biases inherent in medical images and reports.

Medical images, often captured from consistent angles (e.g., frontal), tend to have similar appearances but contain subtle, localized abnormal regions. To better identify these abnormal regions, some studies enhanced the alignment between abnormal regions and corresponding disease tags (You et al., 2021; Liu et al., 2021a), generating disease-grounded visual features. Liu et al. (2021b) introduced a differentiated attention mechanism that subtracts common features from the input image, enabling the model to better focus on abnormal regions. Tanida et al. (2023) utilized a scene graph dataset to detect anatomical regions and describe corresponding abnormal regions, enhancing the explainability of the model. All of these prior studies aimed to overcome the limitations of medical images that are visually biased due to localized abnormal regions. However, none of them have thoroughly analyzed or empirically shown the existence of visual bias.

Medical reports are relatively lengthy, comprising multiple sentences that describe both normal and abnormal findings. Early approaches attempted to generate long reports by integrating relational memory into transformers or incorporating memory matrices. (Chen et al., 2020; 2022). However, these methods often struggled to accurately describe abnormal findings, as they prioritized generating extended narratives over capturing specific abnormalities. To improve the precision of abnormal findings, more recent works have incorporated prior knowledge into MRG models using medical knowledge graphs (Zhang et al., 2020; Liu et al., 2021a; Huang et al., 2023; Li et al., 2023). All of these studies aimed to address so-called textual bias, which has been inconsistently defined— sometimes based on text length and at other times on the articulation of abnormal findings. That is, none of the previous works have provided a clear definition of textual bias.

In this paper, we establish precise definitions for visual bias and textual bias and rigorously confirm the presence of each bias. We believe that this attempt will promote more focused and productive discussions in future MRG research.

## 3 PRELIMINARIES

### 3.1 MEDICAL REPORT GENERATION

Advances in deep learning for computer vision (CV) and natural language processing (NLP) have spurred progress in natural image captioning, which involves generating descriptive text given images (Lin et al., 2014). This success has been extended into the healthcare domain, particularly through medical report generation (MRG). MRG aims to assist radiologists by automatically generating diagnostic reports from medical images. The goal of MRG is not only to ensure accurate disease identification but also to generate context-rich reports.

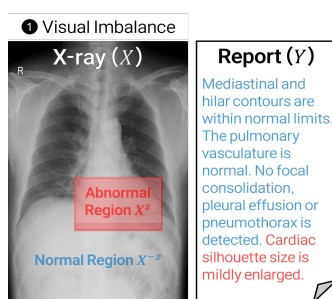 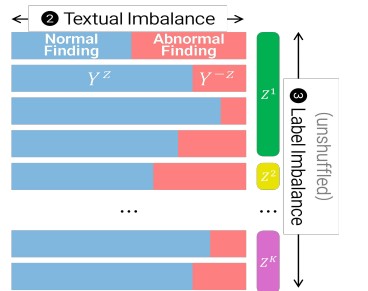 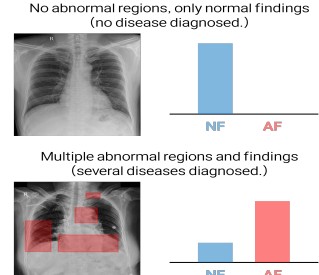

(a) X-ray images include a large normal region with a small abnormal region. The red bounding box is the radiologist annotation.

(b) Each X-ray image is paired with a medical report containing many normal findings (NF) and few abnormal findings (AF).

(c) The upper image showcases a negative sample (i.e., a normal case), while the lower image displays a case with multiple diseases.

Figure 1: Illustration of characteristics in the MRG dataset. (a) presents a typical example of a chest X-ray image, highlighting localized abnormal regions. (b) visualizes the imbalanced ratio of normal to abnormal findings. (c) shows two unusual cases where abnormal findings are absent or abundant.

## 3.2 Terms and Notations

$X \in \mathbb{R}^{W \times H \times C}$ represents a medical image, specifically a chest X-ray as shown in Figure 1a, where $W$, $H$, and $C$ denote the width, height, and number of channels, respectively. Each medical image is paired with a corresponding medical report $Y = [y_1, \cdots, y_t, \cdots, y_T] \in \{0, 1\}^{|v|}$, where $y_t \in \mathbb{N}_0^+$ represents the $t$-th token and $|v|$ indicates the size of vocabulary. The $(X, Y)$-pair is provided along with a disease label $Z \in \{0, 1\}^K$ in which $K - 1$ classes are disease-related and the rest one is a non-disease class (i.e., normal class). Let $X^{(z)}$ and $Y^{(z)}$ represent the abnormal region and finding in the $(X, Y)$-pair, with their respective size and amount denoted by $|X^{(z)}|$ and $|Y^{(z)}|$, while $X^{(-z)}$ and $Y^{(-z)}$ indicate the normal regions and findings, with their respective amount given by $|X^{(-z)}|$ and $|Y^{(-z)}|$. For the positive samples, i.e., $Z|X = 1$ and $Z|Y = 1$, the image and report are defined as $X = X^{(z)} \cup X^{(-z)}$ and $Y = Y^{(z)} \cup Y^{(-z)}$, respectively. For the negative samples, i.e., $Z|X = 0$ and $Z|Y = 0$, each image and report is defined as $X = X^{(-z)}$ and $Y = Y^{(-z)}$, respectively.

## 4 Problem Statement

### 4.1 Three Imbalances and Two Biases in MRG dataset

Figure 1 illustrates three key imbalances in MRG datasets. First, X-ray images are mostly composed of normal regions, with only a small portion representing abnormal areas. This visual imbalance makes model performance heavily dependent on the normal regions, resulting in *visual bias*. Second, medical reports are asymmetrically written, with far more sentences describing normal findings than abnormal ones. This textual imbalance causes model performance to rely on the normal findings, leading to *textual bias*. Finally, the distribution of disease labels is highly skewed; certain diseases are common (e.g., cardiomegaly), while others are relatively rare (e.g., pneumothorax). Such a label imbalance can further deteriorate model performance, but we do not explicitly address it given that mitigating visual and textual biases will inherently resolve this issue. Formal definitions of visual bias and textual bias are provided below.

**Definition 4.1** (Visual Bias). *Let $f_{Z|X}$ denote an image classifier trained to predict a disease label $Z$ given an X-ray image $X$. Given that the classification accuracy is highly sensitive to the size of the abnormal region $|X^{(z)}|$, the model exhibits a bias towards classifying images as normal. This bias arises because normal regions typically represent global patterns, while abnormal ones are local.*

**Definition 4.2** (Textual Bias). *Let $f_{Z|\hat{Y}}$ denote a text classifier trained to predict a disease label $Z$ from a generated report $\hat{Y} \sim G_{Y|X}(\hat{y}_t|\hat{Y}_{1:t-1}, X)$ where $G_{Y|X}$ is a report generator. Given that the classification accuracy is highly sensitive to the number of abnormal findings $|Y^{(z)}|$, the model exhibits a bias towards classifying the generated report as normal. This bias arises because normal findings typically represent global patterns, while abnormal ones are local.*

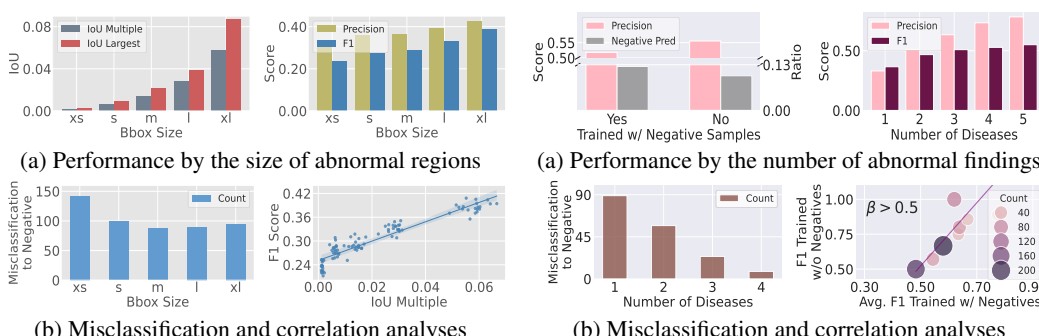

(a) Performance by the size of abnormal regions     (a) Performance by the number of abnormal findings

(b) Misclassification and correlation analyses     (b) Misclassification and correlation analyses

Figure 2: Evidence of visual bias          Figure 3: Evidence of textual bias

## 4.2 EXISTENCE OF VISUAL AND TEXTUAL BIASES

To demonstrate the existence of visual and textual biases, we analyzed IoU and classification accuracy (e.g., precision and F1) in relation to the size of abnormal regions and the number of abnormal findings. The image encoder and text decoder, followed by $f_{Z|X}$ and $f_{Z|\hat{Y}}$, were examined independently to assess the impact of visual and textual biases, respectively. The IoU (Intersection-over-Union) score quantifies the overlap between ground truth bounding boxes and predicted attention regions, as identified by the Grad-CAM heatmap (Selvaraju et al., 2017; Li et al., 2021; Xiao et al., 2023).[1] This metric allows us to evaluate how well the model captures representations relevant to MRG tasks. Classification accuracy measures the performance of the image and text classifiers, $f_{Z|X}$ and $f_{Z|\hat{Y}}$, with values ranging from 0 to 1. A higher score indicates that the image encoder or text decoder has been effectively aligned with MRG tasks.

Figure 2 presents evidence of visual bias. Specifically, Figure 2a shows that as the size of the bounding box (i.e., abnormal region) increases, both IoU and classification accuracy improve. This suggests that as the proportion of normal regions in the X-ray increases, the image classifier $f_{Z|X}$ is more likely to misclassify, indicating that the image encoder is influenced by visual bias. This finding is further supported by Figure 2b. The left plot shows the number of samples misclassified as negative,[2] suggesting that smaller bounding boxes (i.e., larger normal regions) tend to trigger misclassification. The right plot shows a positive correlation between IoU and F1 scores, implying that reduced attention to abnormal regions increases the likelihood of misclassification. This highlights that the abnormal region size contributes to visual bias.

Figure 3 presents evidence of textual bias. The left plot in Figure 3a compares classification accuracy before and after training $G_{Y|X}$, with and without the inclusion of negative samples. The results indicate that the text classifier, $f_{Z|\hat{Y}}$, is more prone to misclassification when negative samples dominate the training data, where the number of abnormal findings is relatively low. The right plot further reinforces this observation: classification accuracy improves as the number of diseases increases. This demonstrates that the number of abnormal findings significantly affects classification accuracy, emphasizing that the text decoder suffers from textual bias.[3] Figure 3b confirms the presence of textual bias. The left plot shows the number of samples misclassified as negative samples, suggesting that fewer abnormal findings are more likely to trigger misclassification. The right plot shows a linear correlation between classification accuracy with and without negative samples, with a slope greater than 0.5.[4][5] This indicates that excluding negative samples improves classification accuracy, further highlighting that the number of abnormal findings is a significant factor contributing to textual bias.

---

[1] See Figure 8 in Appendix A.1.

[2] Negative samples indicate the cases with no documented abnormal regions and findings. The upper image of Figure 1c showcases an example of a negative sample.

[3] More diseases typically correspond to more abnormal findings, making multi-disease cases easier to classify correctly. Refer to the lower image in Figure 1c for an example of a multi-disease case.

[4] For visual clarity, we grouped the F1 predictions by interval along the y-axis, where the bubble size represents the number of predictions in each group.

[5] Note that the x-axis denotes the group-wise average F1 predictions, and a slope greater than 0.5 indicates that the negative samples are imposing a text bias on the model.

### 4.3 FREQUENCY BIAS IN TRANSFORMER ARCHITECTURE

As discussed in previous sections, global patterns, such as normal regions and findings, contribute to visual and textual biases. This bias towards global patterns has been extensively studied from the model's perspective, commonly known as frequency bias or spectral bias. *Frequency bias* refers to the phenomenon where models tend to prioritize low-frequency signals that capture global patterns across multiple samples, while neglecting high-frequency signals that represent local patterns unique to each sample (Schwarz et al., 2021; Tian et al., 2023).

Transformers (Vaswani, 2017) are particularly vulnerable to frequency bias, as the self-attention module functions as a low-pass filter, inherently paying more attention to low-frequencies than high-frequencies (Wang et al., 2022; Park & Kim, 2022; Piao et al., 2024). This globality-seeking behavior of the self-attention module has also been discussed in relation to Principal Component Analysis (PCA) (Zhou et al., 2023; Teo & Nguyen, 2024). Given this, MRG models, where transformers are dominantly used, are especially susceptible to frequency bias, because as described in §4.1 and §4.2, the training data itself is inherently biased towards global patterns. Therefore, mitigating frequency bias is an important and obvious challenge in MRG tasks. The following sections introduce our simple yet powerful approach to addressing this issue.

## 5 METHOD

### 5.1 PRETRAINED ENCODER-DECODER NETWORK

**Vision Transformer for Image Encoder**   Vision Transformer (ViT) (Dosovitskiy, 2020) was the first to successfully apply the transformer architecture directly to image recognition tasks. ViT processes images as sequences of patches, enabling it particularly effective for medical imaging, where abnormalities may span large or subtle regions, such as in X-rays. Accordingly, we implemented an image encoder using the ViT-B model pre-trained on ImageNet (Russakovsky et al., 2015), a widely used approach for medical image encoders. The key ingredient of the ViT encoder is the attention module, encoding each image by aggregating all patchified views. An image embedding, $U \in \mathbb{R}^{N \times |d|}$, processed by a ViT encoder is computed as:

$$U = \text{Attention}(X_p) = \text{softmax}\left(\frac{EW_Q(EW_K)^\text{T}}{\sqrt{d}}\right) EW_V \quad \text{where} \quad E = X_p W_E . \quad (1)$$

Here, $X_p \in \mathbb{R}^{N \times (P^2 \times C)}$ denotes a patchified image sequence with $N$, $P$ and $C$ as the number of patches, the patch size, and the number of channels, respectively. $W_E \in \mathbb{R}^{(P^2 \times C) \times |d|}$ represents the weight matrix mapping each image to the embedding vector. $W_Q \in \mathbb{R}^{|d| \times |d_q|}$, $W_K \in \mathbb{R}^{|d| \times |d_k|}$, $W_V \in \mathbb{R}^{|d| \times |d|}$ are the query, key, and value weights, respectively, and $\sqrt{d}$ is a scaling factor.

**Biomedical GPT for Text Decoder**   Pre-training models on domain-specific data, such as biomedical text, has been shown to significantly enhance downstream task performance (Peng et al., 2019; Lee et al., 2020; Beltagy et al., 2019). Following this approach, we used a biomedical GPT model as the text decoder. Specifically, we initialized the weights of the text decoder based on Papanikolaou & Pierleoni (2020), which fine-tuned the GPT model using biomedical relations extracted from the PubMed corpus. By doing so, the text decoder can better capture domain-specific details or knowledge and is expected to improve the quality and fluency of medical reports accordingly.

**Cross-Attention Module**   The cross-attention module aligns the image embedding with the text embedding. Specifically, the query vector is derived from the text decoder, while the key and value vectors are sourced from the image encoder. As shown in Eq. (2), the cross-attention mechanism computes attention weights, obtained via the softmax($\cdot$) function, by aligning the text embedding $V$ with the image embedding $U$. These attention weights are then used to re-weight the image embedding, allowing the model to aggregate visual features based on their relevance to the textual context. As a result, the aligned representation $A \in \mathbb{R}^{T \times |d|}$ is generated, representing the fused information of both image and text embeddings in a unified space:

$$A = \text{Attention}(U, V) = \text{softmax}\left(\frac{VW_Q(UW_K)^\text{T}}{\sqrt{d}}\right) UW_V . \quad (2)$$

## 5.2 High-Frequency Amplification Layer

As described in §4.3, the attention module introduces an inductive bias, so-called the frequency bias, having transformer-based models less focused on local patterns. To address this, we introduce a *high-frequency amplification layer* (HAL), wherein Fourier transform, high-pass filtering, and inverse Fourier transform are applied subsequently. This layer enhances the model's ability to capture fine-grained details, thereby mitigating its bias towards global patterns.

**Fourier Transform** The Fourier transform decomposes a function into its constituent frequencies using sinusoids as basis functions (Heckbert, 1995). Since both patches and tokens are discrete data, we applied the discrete Fourier Transform (DFT) which is denoted as an operator $\mathcal{T}$:

$$\mathcal{T} : A \to F \quad \text{where} \quad F_c = \sum_{t=0}^{T-1} A_t e^{-\frac{2\pi i}{T} tc} , \quad 0 \le c \le T - 1 .$$

Here, $F_c$ is the $c$-th frequency component, $x_t$ is the $t$-th time-domain signal, and $i$ is the imaginary unit. Computing the DFT directly has a complexity of $O(T^2)$, which is inefficient for large datasets. To overcome this, the Fast Fourier Transform (FFT) was proposed, reducing the complexity to $O(T \log T)$ (Cooley & Tukey, 1965; Brigham, 1988). We apply the FFT to the aligned representation $A \in \mathbb{R}^{T \times |d|}$ using a two-dimensional DFT: one 1D DFT along the time axis, $\mathcal{T}_{\text{time}}$, and another along the feature axis, $\mathcal{T}_{\text{feature}}$, as in (Lee-Thorp et al., 2021; Lee & Lee, 2024). This yields the frequency of the aligned representation denoted as $F \in \mathbb{C}^{T \times |d|}$:

$$F = \mathcal{T} \circ A = \mathcal{T}_{\text{time}}(\mathcal{T}_{\text{feature}}(A)) .$$

**High-Pass Filtering and Inverse FFT** The frequency representation, $F$, consists of low and high frequencies, corresponding to global and local patterns, respectively. In our context, global patterns capture normal regions and findings across $(X, Y)$-pairs, while local patterns represent abnormal ones unique to each pair. High-pass filtering (HPF) is applied to emphasize these local patterns by removing low-frequency components, thus enabling the model to focus on fine-grained details (Pollack, 1948; Costen et al., 1996; Tamkin et al., 2020). Specifically, HPF eliminates frequency components below a certain threshold $\alpha$ by setting $F_{c,d} \leftarrow 0$ for all $c, d \le \alpha$.[6] This operation is implemented using a binary mask $F_{\text{HPF}} = F \odot M$, where $M = \{m_{c,d} \mid m_{c,d} \in \{0, 1\}, 0 \le c \le T - 1, 1 \le d \le |d|\}$, with $m_{c,d} = 1$ for high-frequency components and $m_{c,d} = 0$ otherwise. Finally, the original representation $A$ is reconstructed by transforming $F_{\text{HPF}}$ back to the original domain through an inverse FFT (iFFT):

$$A_{\text{HPF}} = \mathcal{T}^{-1} \circ F_{\text{HPF}} = \mathcal{T}_{\text{feature}}^{-1}(\mathcal{T}_{\text{time}}^{-1}(F_{\text{HPF}})) .$$

## 6 Experimental Setup

**Dataset** We evaluate our model on two widely used medical report generation benchmarks, i.e., MIMIC-CXR and IU X-ray. **1) MIMIC-CXR** is the largest radiography dataset with 377,110 chest X-ray images and 227,827 reports from 65,379 patients. We followed the data split and preprocessing steps from (Chen et al., 2020), and used only frontal view images and reports with more than three tokens, resulting in 153,130 images for the training set, 1,201 for the validation set, and 2,193 for the test set. **2) IU-Xray** is a relatively small public radiography dataset that comprises 7,470 chest X-ray images and 3,955 reports from a total of 3,955 patients. Following the approach of (Chen et al., 2020; Li et al., 2023), we used the dataset only when both frontal and lateral view images were available for each report, resulting in 2,069 images for the training set, 296 for the validation set, and 590 for the test set.

**Baselines** We compare our model with state-of-the-art models on two benchmark datasets. R2Gen (Chen et al., 2020), R2GenCMN (Chen et al., 2022) have been widely used as baseline MRG models.

---

[6]$\alpha$ represents the distance from the origin within the 2D frequency space $F \in \mathbb{C}^{T \times |d|}$. In this domain, components closer to the center represent lower frequencies, while those further from the center represent higher frequencies. Therefore, a lower $\alpha$ removes only a small number of low-frequency components near the origin, whereas a higher $\alpha$ eliminates components up to relatively higher frequencies, farther from the origin.

AlignTransformer (You et al., 2021), CA (Liu et al., 2021b), RGRG (Tanida et al., 2023) are proposed to address visual bias, while PPKED (Liu et al., 2021a), KiUT (Huang et al., 2023), DCL (Li et al., 2023) are designed to address textual bias. Additionally, we include R2GenGPT (Wang et al., 2023b), METransformer (Wang et al., 2023a), and PromptMRG (Jin et al., 2024) which are widely regarded as SOTA models. For the IU X-ray dataset, we include two additional baselines, CVT2Dis (Nicolson et al., 2023), and M2KT (Yang et al., 2023) which have been used for comparing clinical efficacy performance.

**Evaluation Metrics** To measure the fluency and quality of the generated reports, we evaluate them using natural language generation (NLG) metrics, including BLEU (Papineni et al., 2002), METEOR (Denkowski & Lavie, 2011), and ROUGE-L (Lin, 2004).[7] For the clinical efficacy (CE), we include metrics such as precision, recall, and F1. The CheXbert labeling tool (Smit et al., 2020) is used to convert each report into 14 disease classification labels.

**Implementation Details** For the MIMIC-CXR dataset, we use a single frontal view image, while for the IU X-ray dataset, we utilize a pair of images captured from different views of the patient as input. To ensure compatibility across both datasets, all images are resized to 224 and transformed into visual tokens. For the IU X-ray dataset, an additional step is performed where the paired images are concatenated along the embedding dimension and projected back to the original embedding dimension. The hyperparameter $\alpha$ for the HPF is set to 8, and the sensitivity to $\alpha$ is analyzed in Figure 4. We use AdamW optimizer (Loshchilov, 2017) with a learning rate of 5e-6 and a weight decay of 0.05. The learning rate is scheduled using a cosine annealing scheduler, with warm restarts every 5 iterations. The model is trained on an A100 GPU with a batch size of 64 for 39 epochs.

## 7 RESULTS

The results discussed in this section are primarily based on our model trained on the MIMIC-CXR dataset. In §7.3, we provide an additional evaluation conducted on the IU-Xray dataset. We hypothesize that the results will generalize well to other MRG datasets.

### 7.1 GENERALIZATION ASSESSMENT

As discussed in §5.2, HAL reconstructs the original feature representation, $A$, using a limited number of filtered high-frequency components, $F_{HPF}$. Since high-frequency components capture fine-grained local details of the input signals, the reconstructed representation, $A_{HPF}$, may be more prone to overfitting. To assess this risk, we computed the average accuracies and losses and analyzed their trends across both training and validation sets. Figure 4 illustrates the training and validation performance according to different $\alpha$ over 20 epochs. We calculated hit accuracy and categorical cross-entropy loss between the ground truth and predicted tokens. The results indicate that applying HPF with a higher $\alpha$ does not lead to overfitting but consistently improves generalization. We attribute this outcome to the balancing effect between the low-frequency bias inherent in the model and the high-frequency bias introduced by HAL, which allows the model to learn balanced representations that enhance its emergent generalizability. Based on this result, the default setting for the analyses in the following sections is fixed at $\alpha = 8$.

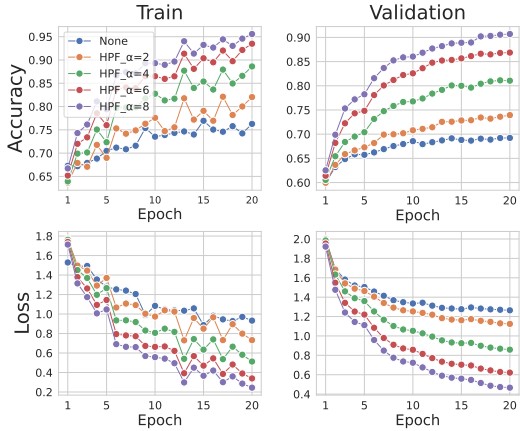

Figure 4: Training and validation performance according to different $\alpha$.

---

[7]https://github.com/tylin/coco-caption

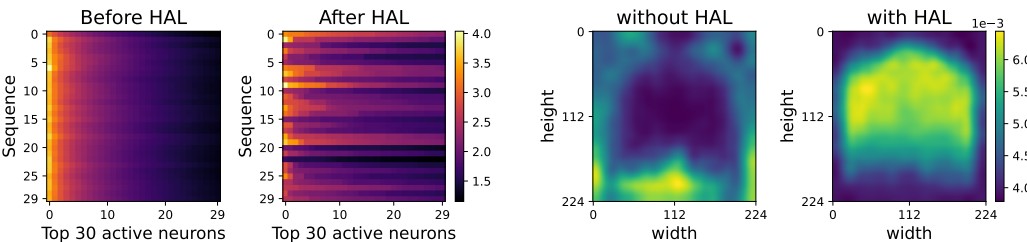

Figure 5: Comparison of neuron activation intensity before and after HAL ($\alpha = 8$)

Figure 6: Comparison of cross-attention map without and with HAL ($\alpha = 8$)

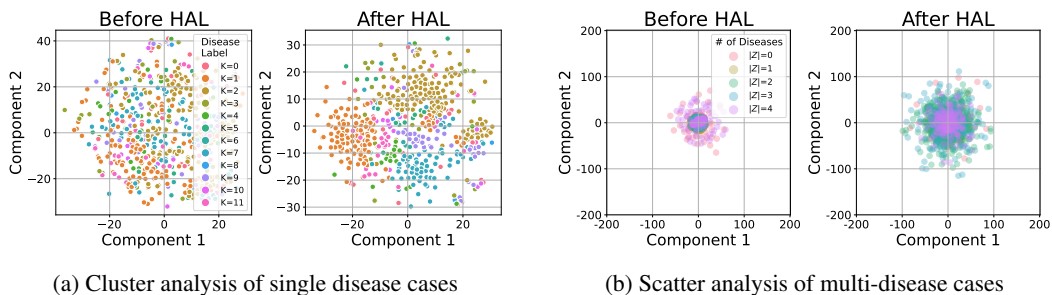

(a) Cluster analysis of single disease cases

(b) Scatter analysis of multi-disease cases

Figure 7: Comparison of T-SNE embeddings before and after HAL ($\alpha = 8$)

## 7.2 Ablation Studies

In this section, we present a comprehensive analysis of the impact of HAL on internal model dynamics, focusing on key aspects such as neuron activation patterns, cross-attention distributions, and representation topology.

**Pseudo-spectrogram Analysis**  A spectrogram provides a visual representation of how frequency components evolve over time, typically depicting frequency on the x-axis and time on the y-axis, with color indicating the intensity of each frequency component. Inspired by this approach, we conducted a pseudo-spectrogram analysis of neuron activation. Figure 5 compares the neuron activation intensity before and after HAL, where the x-axis represents the top 30 neurons ranked by activation level and the y-axis denotes the temporal sequence of tokens. The figure shows that in the layer before HAL, only a subset of neurons are strongly activated, with most neurons remaining inactive. Furthermore, those few active neurons exhibit uniform activation across all tokens in the sequence, suggesting an indistinguishable activation pattern. In contrast, there is a rich and non-uniform activation after HAL. An activation spectrum indicates that HAL allows the model to effectively capture fine-grained details by amplifying high-frequency signals, which might otherwise be ignored. Consequently, HAL produces a richer representation so that neuron activation forms a spectrum, ultimately improving the model's discriminative perceptiveness.

**Cross-attention Analysis**  HAL is placed after the cross-attention layer, making it highly dependent on the influence of HAL. Therefore, comparing the cross-attention map with and without HAL helps illustrate how it has affected the image-to-text alignment. Figure 6 shows a comparison of the cross-attention distributions across ($224 \times 224$) images for models trained with and without HAL. The results reveal clear advantages of using HAL. In the model without HAL, cross-attention tends to focus on the periphery, especially the mid-abdominal part, which contains little information about chest disease. This may be due to the "common" appearance of grey areas around the abdomen on most X-ray images (see Figures 1 and 8), representing a global pattern across all samples regardless of disease type. On the other hand, the model trained with HAL shows that the image-to-text cross-attention is concentrated around the center of the image (i.e., the upper-mid-thoracic region), typically containing the "specific" information of chest disease. This may be understood as evidence

Table 1: The comparison of model performance on MIMIC-CXR dataset. Note that **bold** numbers highlight the best performance, underlined numbers indicate the second-best performance, and asterisked ($*$) numbers denote the third-best performance, respectively.

| Baselines | NLG Metrics | | | | | | CE Metrics | | |
|---|---|---|---|---|---|---|---|---|---|
| | BLEU-1 | BLEU-2 | BLEU-3 | BLEU-4 | METEOR | ROUGE-L | Precision | Recall | F1 |
| R2Gen | 0.353 | 0.218 | 0.145 | 0.103 | 0.142 | 0.277 | 0.333 | 0.273 | 0.276 |
| R2GenCMN | 0.353 | 0.218 | 0.148 | 0.106 | 0.142 | 0.278 | 0.334 | 0.275 | 0.278 |
| AlignTransformer | 0.378 | 0.235 | 0.156 | 0.112 | 0.158 | 0.283 | - | - | - |
| CA | 0.350 | 0.219 | 0.152 | 0.109 | 0.151 | 0.283 | 0.352 | 0.298 | 0.303 |
| RGRG | 0.373 | 0.249 | 0.175* | 0.126* | 0.168 | 0.264 | 0.461* | 0.475 | 0.447 |
| PPKED | 0.360 | 0.224 | 0.149 | 0.106 | 0.149 | 0.284 | - | - | - |
| KiUT | 0.393 | 0.243 | 0.159 | 0.113 | 0.160* | 0.285 | 0.371 | 0.318 | 0.321 |
| DCL | - | - | - | 0.109 | 0.150 | 0.284 | 0.471 | 0.352 | 0.373 |
| R2GenGPT | **0.411** | **0.267** | 0.186 | 0.134 | 0.160* | 0.297 | 0.392 | 0.387 | 0.389 |
| METransformer | 0.386 | 0.250* | 0.169 | 0.124 | 0.152 | 0.291* | 0.364 | 0.309 | 0.311 |
| PromptMRG | 0.398* | - | - | 0.112 | 0.157 | 0.268 | **0.501** | **0.509** | **0.476** |
| Ours (HAL) | 0.399 | 0.264 | **0.189** | **0.143** | **0.170** | **0.299** | 0.434 | 0.410* | 0.392* |

Table 2: The comparison of model performance on IU-Xray dataset. Note that **bold** numbers highlight the best performance, underlined numbers indicate the second-best performance, and asterisked ($*$) numbers denote the third-best performance, respectively.

| Baselines | NLG Metrics | | | | | | CE Metrics | | |
|---|---|---|---|---|---|---|---|---|---|
| | BLEU-1 | BLEU-2 | BLEU-3 | BLEU-4 | METEOR | ROUGE-L | Precision | Recall | F1 |
| R2Gen | 0.470 | 0.304 | 0.219 | 0.165 | 0.187 | 0.371 | 0.141 | 0.136 | 0.136 |
| R2GenCMN | 0.475 | 0.309 | 0.222 | 0.170 | 0.191 | 0.375 | - | - | - |
| AlignTransformer | 0.484 | 0.313 | 0.225 | 0.173 | 0.204 | 0.379 | - | - | - |
| CA | 0.492 | 0.314 | 0.222 | 0.169 | 0.193 | 0.381 | - | - | - |
| RGRG | - | - | - | - | - | - | 0.183* | 0.187* | 0.180* |
| PPKED | 0.483 | 0.315 | 0.224 | 0.168 | 0.190 | 0.376 | - | - | - |
| KiUT | **0.525** | 0.360 | 0.251 | 0.185 | 0.242 | 0.409 | - | - | - |
| DCL | - | - | - | 0.163 | 0.193 | 0.383 | 0.168 | 0.167 | 0.162 |
| R2GenGPT | 0.488 | 0.316 | 0.228 | 0.173 | 0.211* | 0.377 | - | - | - |
| METransformer | 0.483 | 0.322* | 0.228 | 0.172 | 0.192 | 0.38 | - | - | - |
| PromptMRG | 0.401 | - | - | 0.098 | 0.160 | 0.281 | 0.213 | 0.229 | 0.211 |
| CVT2Dis | 0.473 | 0.304 | 0.224 | 0.175* | 0.200 | 0.376 | 0.174 | 0.172 | 0.168 |
| M2KT | 0.497* | 0.319 | 0.230* | 0.174 | - | 0.399* | 0.153 | 0.145 | 0.145 |
| Ours (HAL) | 0.521 | **0.425** | **0.371** | **0.336** | **0.263** | **0.507** | **0.418** | **0.415** | **0.414** |

that HAL enhances robustness to the frequency bias. In summary, Figure 6 demonstrates that HAL improved the model to attend to diagnostically significant regions by mitigating the frequency bias.

**Representation Analysis**   Comparing the topology of representation before and after a specific layer provides an intuitive explanation of how it works as an operator, and proves the utility it yields from the perspective of representation quality. In this regard, we performed the T-SNE embedding (Van der Maaten & Hinton, 2008) and visualized representation for both single-disease and multi-disease cases, as shown in Figure 7. For single-disease cases (see Figure 7a), which encompass 12 distinct diseases, the embedding vector before HAL produces entangled clusters, indicating poor feature discrimination by diseases. In contrast, the embeddings after HAL form well-separated clusters, suggesting a marked improvement in representation quality. This improvement is likely due to HAL, where amplified high-frequency signals highlight the local patterns unique to each sample, but erase the global patterns shared across samples, which contribute to entangled representations.

For multi-disease cases (see Figure 7b), we cannot conduct cluster analysis as T-SNE embeddings fail to build distinguishable representations due to the high complexity of disease combinations—the complex nature of these combinations results in highly entangled feature representations—making it challenging to achieve well-separated clusters.[8] Instead, we can do scatter analysis to demonstrate whether HAL makes a dispersed representation—the larger dispersion means that the model treats each point more uniquely. Before HAL, the T-SNE embeddings show compact representation, except for the $|Z| = 4$ case that exhibits dispersed representation. This dispersion is likely due to the

---

[8]In multi-disease cases, there are many samples that have the same disease in common. In these cases, the overlapping diseases among samples dilute or mix the distinctive local patterns. That is, the local patterns become nothing but noise, and only the global patterns survive. As a result, the frequency bias in models becomes more pronounced compared to single-disease scenarios, making it challenging to do cluster analysis.

reduced influence of normal findings, as illustrated in Figure 1c, enabling clearer differentiation between samples. After HAL, the embeddings appear to spread more dispersely, implying that each sample is embedded finely enough to be distinguishable. That is, the scatter analysis suggests that HAL increases the model's perceptiveness to the local details and thus mitigates frequency bias.

### 7.3 PERFORMANCE COMPARISON

Through the previous sections has it been shown that HAL is an effective tool for addressing frequency bias, especially crucial in MRG tasks where visual and textual biases are already prevalent. Nevertheless, one might argue that HAL, due to its simplicity, may not be competitive against existing methods for MRG. To address this concern, this section presents a comparative evaluation on two MRG benchmark datasets: MIMIC-CXR and IU-Xray. Tables 1 and 2 summarize the performance of MRG models using both NLG and CE metrics, illustrating that our model performs competitively against other baselines. Specifically, our model outperforms baseline models on NLG metrics, demonstrating its ability to generate high-quality reports containing featured medical terminology found in real-world clinical texts. Furthermore, when comparing the top-3 ranks for each metric, our model ranks consistently high across almost all metrics. This balanced achievement across diverse metrics suggests that HAL not only enhances the overall quality of generated reports but also provides robustness in capturing key clinical concepts, making it a reliable tool for MRG tasks. Note that the performance of baselines was taken directly from the results reported in the original papers.[9]

## 8 LIMITATIONS AND FUTURE WORK

It is important to note that our current results were obtained without extensive hyperparameter optimization. We believe that a systematic exploration of hyperparameters could further enhance the model's performance and stability, providing stronger evidence of HAL's effectiveness. The primary goal of HAL is to reduce the impact of global patterns by amplifying high-frequency signals. However, this approach may be less effective if the training data is either insufficient or contains too much randomness, making it difficult for dominant global patterns to emerge. In such cases, HAL might even introduce a bias towards local patterns instead. Accordingly, future research should explore methods to balance global and local patterns, especially when training data is limited or noisy. In addition, while this study empirically links visual and textual biases with frequency bias, additional theoretical grounding is needed to strengthen those empirical findings. We hope that future research will further explore this area. Meanwhile, further improvements could also be explored in the pre-training phase. We anticipate that pre-training the encoder or decoder models on chest X-ray data would yield greater performance.

## 9 CONCLUSION

In this work, we demonstrated the existence of visual and textual biases in the MRG dataset (§4.1) and discussed how these biases make MRG models especially prone to frequency bias, with a tendency to prioritize low-frequency components. To counter this vulnerability, we introduced the *high-frequency amplification layer* (HAL) (§5.2), designed to mitigate the model's predisposition towards such biases. Our results showed that HAL significantly enhances various aspects, including neuron activation, cross-attention map, and representation quality, as detailed in ablation studies (§7.2). Despite its simplicity, HAL exhibited outstanding performance in comparative evaluations (§7.3). All these findings strongly support our arguments that: (1) debiasing is a fundamental issue for improving MRG tasks, and (2) mitigating frequency bias is crucial for enabling models to capture both global and local patterns of medical images more effectively. We believe this work will pave the way for more robust MRG models that are better equipped to handle the complexities of real-world medical data, ultimately contributing to advanced medical imaging tasks.

---

[9]It is important to note that only Jin et al. (2024) reported CE performance on the IU-Xray dataset. Therefore, the CE metrics presented in Table 2 are all borrowed from the results reported in Jin et al. (2024). Unlike the NLG metrics, the CE metrics were evaluated in a zero-shot setting, where the models were not trained on the IU-Xray dataset. This context explains why the baseline models exhibit relatively lower CE performance compared to our model.

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

# A APPENDIX

## A.1 VISUAL BIAS EXPERIMENTS

According to existing studies (Liu et al., 2021b; You et al., 2021; Tanida et al., 2023), localized abnormal regions are difficult to capture. In this experiment, we investigated several potential factors that may affect visual bias, including the bounding box size. We used VinDr-CXR dataset which provides 18,000 chest X-ray images annotated with bounding box information for disease regions and 23 disease labels.[10] We utilized ViT-S pre-trained on 510K X-ray images for this section (Xiao et al., 2023). To evaluate the model's ability to capture disease-relevant regions, we employed IoU (Intersection-over-Union) metric. Specifically, we utilized the IoU Multiple metric, which compares all actual bounding boxes to all predicted bounding boxes, and the IoU Largest metric, which compares the largest actual bounding box and the largest predicted bounding box. Classification performance was assessed using precision and F1.[11]

**Grad-CAM in ViTs** In this paragraph, we briefly explain how we calculated the IoU score based on Grad-CAM. Grad-CAM (Selvaraju et al., 2017) is a method for generating visual explanations to identify which parts of the input image have the most influence on a given class prediction. It was originally proposed for CNN architecture and is now applicable to ViTs as well. Similar to CNN-based models, which extract feature maps from the last convolutional layer, Grad-CAM for ViTs extracts feature maps from the norm1 layer of the final block. These feature maps have the shape of $f \in \mathbb{R}^{N \times (P^2 \times C)}$, where $N$, $P$, and $C$ represent the number of patches, the patch size, and the number of channels, respectively. The feature maps are then projected back into the original image space by reversing the patchification process, allowing for spatial interpretation. In this experiment, we extracted activation maps based on the ground truth disease label. To generate predicted bounding boxes, we retained only the regions of the activation maps that exceed 75% of the maximum activation value. We then utilized OpenCV's findContours function to detect the contours of these regions, followed by the minAreaRect function to generate the minimum area bounding rectangles that enclose each contour (see Figure 8).

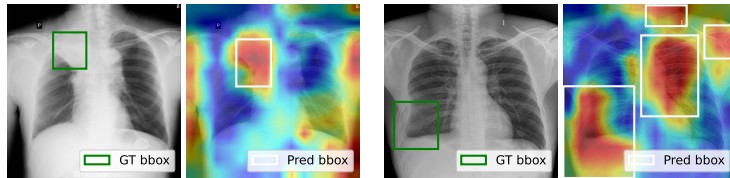

Figure 8: Bounding box generation from Grad-CAM visualizations

**Results** In §4.2, we demonstrated that as the **(1) bounding box size** (i.e., abnormal regions) decreases, both the ability to capture abnormal regions and accuracy of disease classification deteriorate. Additionally, the number of positive samples misclassified as negative increases, highlighting the presence of visual bias. We then analyzed the effect of **(2) biased distribution of diseases** on the model performance. In Figure 9a, the IoU and classification scores of the vanilla classification model for different diseases are sorted in ascending order based on the number of training samples. Notably, disease ID 22 represents the negative samples. While classification performance tends to improve with an increasing number of training samples, the IoU score does not consistently reflect the classification accuracy. To examine potential correlations, we plotted a regression using bootstrapped samples, ensuring a minimum of 100 samples per class and down-weighting outliers. As shown in Figure 9b, the regression plot reveals no significant correlation between IoU and classification performance. The above analysis is consistently observed when comparing performance by the **(3) number of diseases** per image, as shown in Figure 10. For cases where medical images contain multiple diseases (excluding classes with fewer than 30 samples), the results show that as the number of diseases per sample increases, the classification performance improves, while the IoU

---

[10]https://physionet.org/content/vindr-cxr/1.0.0/

[11]In this analysis, precision is calculated by considering only the top-k predicted labels as the predicted disease, where k corresponds to the number of ground truth disease labels for each sample.

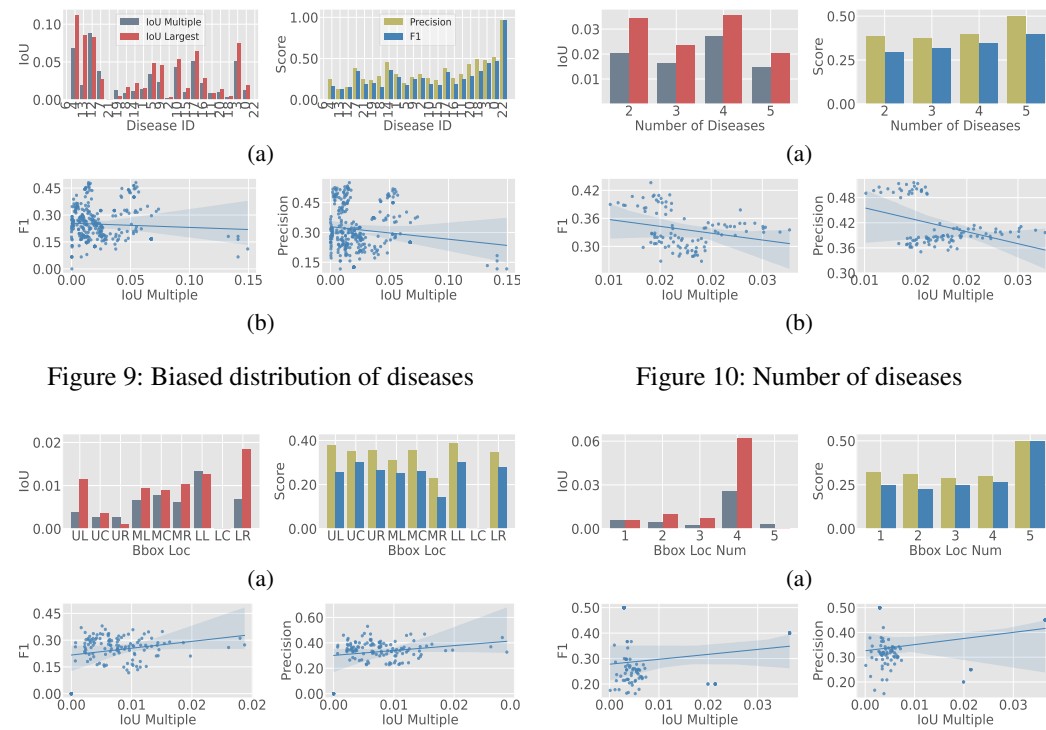

Figure 9: Biased distribution of diseases

Figure 10: Number of diseases

Figure 11: Location of bounding boxes

Figure 12: Scatterness of bounding boxes

scores remain unaffected. We further analyzed whether the **(4) location of bounding box** might influence the model performance. The image is divided into nine areas ($3 \times 3$ grid) where the y-axis represents upper, middle, and lower regions, and the x-axis represents left, center, and right regions. In Figure 11b, a weak positive correlation is observed due to outliers, but in Figure 11a, the upper center (UC), middle center (MC), and lower right (LR) areas show significant differences in IoU performance, despite having nearly identical classification performance. This implies there is little correlation between these two metrics. Additionally, no significant patterns were identified when analyzing the **(5) scatterness of bounding box** based on the number of differently located bounding boxes, as shown in Figure 12. This experiment confirmed that the primary factor influencing visual bias is the size of the bounding box.

## A.2 TEXTUAL BIAS EXPERIMENTS

In this experiment, we investigated several potential factors that may contribute to textual bias in MRG, including the number of abnormal findings. We used the MIMIC-CXR dataset and the baseline MRG model without any HPF. We prepared ground truth labels using CheXbert labeling tool (Smit et al., 2020) for the analysis, but the "support devices" label was excluded as it does not represent an actual disease. The results were evaluated using precision and F1 score for CE metrics, and BLEU-4, METEOR, and ROUGE-L for NLG metrics.

**Results** In §4.2, we demonstrate that as the **(1) number of abnormal findings** decreases, the model's diagnostic performance degrades. In other words, the model exhibits a textual bias towards dominant normal findings. This is further supported by the increasing misclassification tendency as negative samples dominate the training data. Additionally, we analyzed the effect of **(2) biased distribution of diseases**. As shown in Figure 13a, the NLG and CE scores of MRG model for different diseases are sorted in ascending order based on the number of training samples, with disease label 13 representing negative samples. In contrast to the visual bias experiment, both NLG and CE metrics showed no significant differences for abnormal diseases, except in the case of negative samples. Additionally, Figure 13b does not appear to have a strong correlation. Next, we analyzed

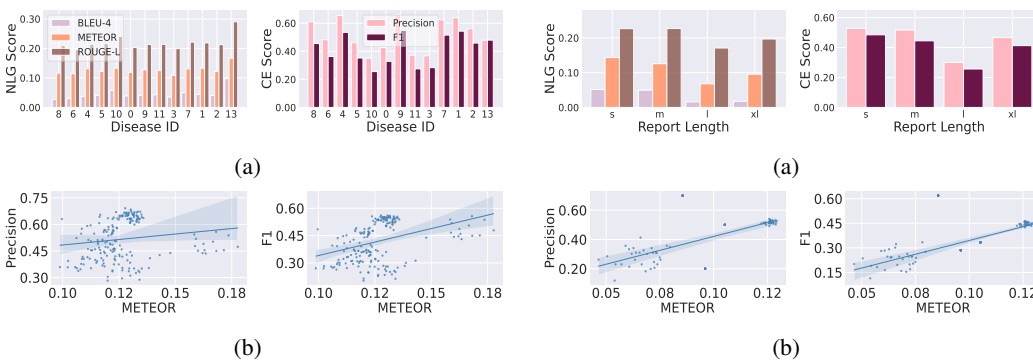

Figure 13: Biased distribution of diseases

Figure 14: Length of reports

the impact of the **(3) report length**, categorizing reports as short (20 words or less), medium (up to 50 words), long (up to 80 words), and extra long (more than 80 words) given the max length is 100. As shown in Figure 14a, short to medium-length generated reports tend to perform better on both NLG and CE metrics. However, the scores do not show any clear upward or downward trend based on the generated length, suggesting that the results may not be statistically significant. Figure 14b shows a positive correlation between the NLG score and CE score. This experiment confirmed that the primary factor influencing textual bias is the number of abnormal findings.

### A.3 VIT OUTPERFORMS RESNET IN MEDICAL IMAGING TASKS

Convolutional Neural Networks (CNNs) have been widely used in various computer vision tasks. After the advent of Vision Transformers (ViTs), ViTs have shown its potential as a competitive alternative to CNNs. Since CNNs and ViTs each exhibit distinct advantages and limitations, it is general to choose the appropriate backbone model based on the downstream tasks. For instance, CNNs possess a high inductive bias, while ViTs effectively capture long-range dependencies. Although CNN-based models might seem suitable for the localized nature of abnormalities in medical images, numerous studies have demonstrated the effectiveness of ViTs in automated medical image diagnosis, ranging from medical image segmentation (Karimi et al., 2021), medical image classification (Matsoukas et al., 2021), and medical image reconstruction (Zhang et al., 2021), especially when pre-trained on ImageNet. Since diagnosis often requires consideration of distant organs and tissues, the long-range dependencies captured by ViTs are particularly advantageous in the medical domain. In medical images, abnormal regions may span large or subtle regions. Unlike CNNs, which focus on local features in the lower layers and global features in the higher layers, ViTs capture both local and global features at every layer, preserving fine-grained details and contextual relationships (Raghu et al., 2021). Additionally, sparse attention mechanisms in ViTs are known to enhance robustness against noise (Zhou et al., 2022). ViTs perform particularly well when pre-trained on large-scale datasets or via self-supervision.

This finding is further validated through our simple experiment. Specifically, we compared the classification performance of ResNet-50 (He et al., 2016) and ViT-S (Dosovitskiy, 2020), both pre-trained on ImageNet and fine-tuned on the Vindr-CXR dataset for 50 epochs. The results demonstrated that ViT-S outperformed ResNet-50, despite the latter having a slightly higher number of parameters (See Table 3).

Table 3: Comparison of image encoder backbone models

| Model | Params | F1 | Hit@k |
|---|---|---|---|
| ResNet-50 | 23.9M | 0.682 | 0.546 |
| ViT-S | 22.2M | 0.699 | 0.624 |

