# OpenReview forum: "Debiased Medical Report Generation with High-Frequency Amplification"
_ICLR.cc/2025/Conference — ICLR 2025 Conference Withdrawn Submission_

### Official Review · Reviewer_5KKC · 2024-10-28

**Soundness:** 2
**Presentation:** 2
**Contribution:** 2
**Rating:** 3
**Confidence:** 3

**Summary:**

This paper identifies the challenge of visual and textual biases in automated medical report generation, which stems from the overwhelming presence of normal features in both medical images and reports. The authors define visual bias and textual bias, associating these biases with *frequency bias*, where models tend to emphasize low-frequency (normal) signals over high-frequency (abnormal) signals. To counter this, they propose the High-Frequency Amplification Layer (HAL), designed to heighten the model’s sensitivity to abnormal (high-frequency) details, thus enhancing diagnostic accuracy. Validation on MIMIC-CXR and IU X-ray benchmarks shows HAL’s effectiveness through various analyses and demonstrates competitive or superior performance compared to state-of-the-art models.

**Strengths:**

1. The paper defines visual and textual biases, highlighting their impact on MRG model performance.
2. The empirical analysis of visual and textual biases confirms the presence of each bias and demonstrates the existence of the frequency bias.
3. The work introduces a high-frequency amplification layer to amplify high-frequency signals, enabling improved detection of abnormal features.
4. The paper provides a thorough experimental analysis, including ablation studies and qualitative comparisons, to substantiate the effectiveness of the proposed methods.

**Weaknesses:**

1. Many of the experiments are conducted to demonstrate the presence of visual and textual biases, but the experimental details are not clearly articulated. For example, how many samples were utilized to analyze the visual bias, and what is the text classifier? Moreover, most of the figures for analysis should have more explanations (e.g., Figure 3, Figure 9 and Figure 10). It is not clear right now.
2. It is not clear where HAL applied. Were they adopted in all cross-attention layers?
3. The paper lacks sufficient novelty, as it only combines HAL. However, it does not adequately explain the results of the baseline model.  Exactly how HAL works in the final report generation should be further enhanced with examples of generated reports.

**Questions:**

1. There should be more explanations about why clinical efficacy is lower than the SOTA model. The paper aims at utilizing HAL to capture more abnormal regions, but the CE metric for detecting abnormalities was not been improved.
2. Can HAL be applied in other existing MRG models? More ablation studies of HAL should be conducted to demonstrate its effectiveness.

---

### Official Review · Reviewer_Sqbf · 2024-11-01

**Soundness:** 2
**Presentation:** 3
**Contribution:** 3
**Rating:** 6
**Confidence:** 3

**Summary:**

The paper identifies transformer models’ bias towards low-frequency regions of an image as a potential source for their low performance on the medical report generation (MRG) task. It provides evidence for visual and textual bias, where larger abnormal regions and the number of diseases in a study leads to a higher F1 score of the generated report. Since most of an image is normal, models are biased towards classifying images as normal. The paper addresses this issue by proposing a high-frequency amplification layer (HAL) in order to filter out low-frequency regions. It demonstrates that models trained with HAL learn more discriminative representations of diseases, among other benefits, which leads to comparable performance on natural language generation (NLG) and clinical efficacy (CE) metrics to the state-of-the-art (SoTA).

**Strengths:**

The work presents a novel perspective on medical report generation, identifying bias towards low-frequency regions as a challenge for learning good visual representations. The authors introduce the problem with clarity, providing evidence for the correlation between signal frequency and performance. Using Fourier transforms to filter out low-frequency regions from an image is an interesting solution to the problem. The efficacy of the method is backed by empirical results: the model trains better and achieves comparable performance with the SoTA. Overall, there seems to be potential both for mitigating visual and textual bias as an area of research and this specific method for doing so.

**Weaknesses:**

Table 1:

Although HAL achieves performance that puts it in the top 3, ultimately its F1 is still more than 8 points lower than the SoTA, which brings into question its advantage over PromptMRG. It would be interesting if the performance gain from HAL composes with gains from other methods. For instance, would RGRG + HAL result in better performance than just RGRG alone? Therefore, the authors should include a comparison with a simple transformer that does not use HAL in order to quantify the effect of HAL on model performance.

Table 2:

As the authors themselves noted, this comparison is unfair because the baseline models were evaluated zero-shot on IU-Xray while HAL was trained. The authors should provide a fairer comparison, perhaps by also evaluate zero-shot a model with HAL trained on MIMIC-CXR but not IU -Xray.

Line 130-131:

> Each medical image is paired with a corresponding medical report… indicates the size of the vocabulary.

The notation is weird here. Why does $Y = [y_1, \cdots, y_t, \cdots, y_T]$ belong in $\{0, 1\}^{|v|}$? What does this set, $\{0, 1\}^{|v|}$ refer to?

Line 133-138:

I think it is unnecessary to use math notation here to talk about positive and negative samples. It does not add clarity to the explanation. For example, the notation $|X^{(z)}|$ does not give the reader any more information about how the size of an abnormal region is calculated.

Figure 3a (left) is very hard to read. I cannot figure out which bar has the score of 0.50. Furthermore, although the discussion of textual bias is interesting, it is left unaddressed by the paper as it focuses on visual bias.

**Questions:**

I have listed some questions in the “Weaknesses” section. Below are a few more.

Line 267:

Why is $T$ the first dimension of $A$? I think it should be $N$ because the $U$ is $N \times |d|$.

Line 412 and figure 5:

It is unclear what “neurons” refer to here. Is it the output of the attention layer or MLP layer, or something else?

Figure 4:
How is accuracy calculated here?
Since the loss keeps decreasing for larger $\alpha$, have the authors considered increasing the $\alpha$ beyond 8?

---

### Official Review · Reviewer_APMA · 2024-11-01

**Soundness:** 3
**Presentation:** 2
**Contribution:** 2
**Rating:** 3
**Confidence:** 4

**Summary:**

The authors conduct an examination of visual and textual biases in medical report generation (MRG) datasets. The analysis find that global patterns, such as normal regions and findings, contribute to visual and textual biases. These biases make MRG models prone to frequency bias, where global patterns are prioritized and local patterns (e.g. abnormal findings) are ignored. In order to mitigate this issue, the authors propose an architectural modification in the form of a high-frequency amplification layer (HAL), which aims to enhance a model’s perceptiveness to fine-grained details. HAL reduces biases, leading to improved performance in MRG tasks.

**Strengths:**

1. The paper introduces an approach for improving the quality of medical report generation, which is a high-impact problem with potential for positively impacting the field of medicine.
2. The authors demonstrate that their proposed approach HAL leads to performance improvements over several existing methods in this domain.

**Weaknesses:**

1. **Inadequate evaluations for demonstrating the utility of HAL**: The key claim of this paper is that the proposed method HAL improves robustness of medical report generation models, which may struggle to learn fine-grained abnormal findings. However, this claim is not sufficiently evaluated in Section 7, and as a result, it is unclear if HAL is improving robustness to these biases. Only aggregate performance values are reported on the MIMIC-CXR and IU datasets.

    a. Does HAL improve report generation performance (i.e. NLG and CE scores) when there is a single abnormality in the image? What about multiple abnormalities? Does HAL improve report generation performance when findings are small in size? Does HAL reduce performance on normal cases? All of these questions are critical for determining whether HAL mitigates biases as claimed, but none of these are evaluated.

    b. Additionally, in order to demonstrate the usefulness of HAL, Tables 1 and 2 could benefit from an additional ablation using the exact same experimental setup but without the novel HAL layer.

    c. In Tables 1 and 2, I recommend that the authors use more recently-developed (standard) report generation metrics for evaluating report quality with respect to factuality, such as RadGraph-F1 [1] or RadCliQ [2].

2. **Inadequate evaluations for demonstrating the existence of visual and textual bias in report generation datasets:** The evaluations in Section 4.1 show that a classifier $f_{Z|X}$ trained on the images demonstrates lower performance when abnormalities occupy small regions. Similarly, a classifier $f_{Z|\hat{Y}}$ trained on  generated reports demonstrates lower performance when there are more normal samples in the training data. These results show that the classifier $f$ picks up on several biases, but how do these experiments relate to the report generation task that is the focus of this work? Do report generation models learn these same biases? It is unclear to me why classification models are the focus of this analysis.

3. **Presentation issues:** There are several presentation issues in this manuscript.

    a. First, the notation provided in Section 3.2 is overly convoluted and unclear; for instance, how can the value of an image or text report be set to 0 or 1 (Lines 136-138)? What is meant by positive and negative in this context? This notation also seems unnecessary, since most of this notation is never referenced again in the manuscript.

    b. Additionally, section 4.2 is critical to this paper yet does not include adequate implementation details in the main text to understand the goals of the experiments, with most of this material being relegated to the appendix instead. For instance, details on the classification task, classification model, dataset, etc. are not provided in the main text, making it difficult to understand the problem setup.

[1] Delbrouck et al. "Improving the Factual Correctness of Radiology Report Generation with Semantic Rewards”.” 2022.

[2] Yu et al. “Evaluating progress in automatic chest X-ray radiology report generation.” 2023.

**Questions:**

My questions are listed above in the “weaknesses” section.

---

### Official Review · Reviewer_dpc6 · 2024-11-03

**Soundness:** 3
**Presentation:** 3
**Contribution:** 2
**Rating:** 6
**Confidence:** 3

**Summary:**

This paper addresses the issue of low frequency (global) visual and textual biases in report generation caused by visual imbalance, textual imbalance, and skewed distribution of disease labels. The authors propose a novel approach called the High-Frequency Amplification Layer (HAL), which uses DFFT on the time axis and feature axis, followed by masking to perform high-pass filtering. This method emphasizes high-frequency components in the feature and may reduce biases. The paper includes experiments on the MIMIC-CXR and IV X-ray datasets to demonstrate improved performance.

**Strengths:**

- This paper provides novel insights into visual and textual biases important for report generation and attempts to reduce them.
- It offers a thorough examination of these biases and their impact on model performance.
- The clear problem definition contributes to a more focused discussion in the MRG field.

**Weaknesses:**

- Some experimental settings are unclear; further explanation would improve clarity.
- Although the implementation of the HAL layer (DFFT on the time and feature axes) is simple, this layer requires further comprehensive evaluation and ablation studies.

**Questions:**

1. The structure of the whole model of proposed approach is not clearly mentioned, like how many HAL layers were inserted. Providing a global view of the model pipeline will make it clearer and easier to follow.
2. In line 422, HAL is placed after the cross-attention layer. If HAL is after this layer, how does it influence the already computed cross-attention?
3. In line 201 and Figure 3a, "classification accuracy improves as the number of diseases increases." How should this conclusion be interpreted, given that a higher number of diseases might exacerbate distribution imbalance?
4. How is the hyperparameter alpha for the high-pass filter set to 8? From Figure 4, performance appears to still improve as alpha increases.
5. How was the decision made to train for 39 epochs, while the generalization assessment plots the training/validation curve for 20 epochs?
6. The baseline without the HAL layer is not reported, which could illustrate the influence of the HAL layer on the model.

---

### Note · Authors · 2024-11-25

**Comment:**

Thank you for your thoughtful and constructive feedback on our paper. During the rebuttal process, we realized there are several ways we can improve our work. To make these improvements, we have decided to withdraw our submission.

We deeply appreciate your comments, which will help us make the research stronger.

**Withdrawal Confirmation:**

I have read and agree with the venue's withdrawal policy on behalf of myself and my co-authors.